# Feasibility Study on the Use of Infrared Cameras for Skin Cancer Detection under a Proposed Data Degradation Model

**DOI:** 10.3390/s24165152

**Published:** 2024-08-09

**Authors:** Ricardo F. Soto, Sebastián E. Godoy

**Affiliations:** Departamento de Ingeniería Eléctrica, Facultad de Ingeniería, Universidad de Concepción, Concepción 4070409, Chile; ricsoto@udec.cl

**Keywords:** infrared thermography, skin cancer screening, data adaptation, microbolometer technology

## Abstract

Infrared thermography is considered a useful technique for diagnosing several skin pathologies but it has not been widely adopted mainly due to its high cost. Here, we investigate the feasibility of using low-cost infrared cameras with microbolometer technology for detecting skin cancer. For this purpose, we collected infrared data from volunteer subjects using a high-cost/high-quality infrared camera. We propose a degradation model to assess the use of lower-cost imagers in such a task. The degradation model was validated by mimicking video acquisition with the low-cost cameras, using data originally captured with a medium-cost camera. The outcome of the proposed model was then compared with the infrared video obtained with actual cameras, achieving an average Pearson correlation coefficient of more than 0.9271. Therefore, the model successfully transfers the behavior of cameras with poorer characteristics to videos acquired with higher-quality cameras. Using the proposed model, we simulated the acquisition of patient data with three different lower-cost cameras, namely, Xenics Gobi-640, Opgal Therm-App, and Seek Thermal CompactPRO. The degraded data were used to evaluate the performance of a skin cancer detection algorithm. The Xenics and Opgal cameras achieved accuracies of 84.33% and 84.20%, respectively, and sensitivities of 83.03% and 83.23%, respectively. These values closely matched those from the non-degraded data, indicating that employing these lower-cost cameras is appropriate for skin cancer detection. The Seek camera achieved an accuracy of 82.13% and a sensitivity of 79.77%. Based on these results, we conclude that this camera is appropriate for less critical applications.

## 1. Introduction

Temperature is a key indicator for identifying anomalies in both living and inert systems. Infrared thermography (IRT) is a mature and widely accepted technique used as a non-contact temperature monitoring tool; it is used in the early detection of equipment failures and process anomalies in industrial operations [1], as well as in medical diagnoses [2].

Concerning medical applications of IRT, its use in diagnosing breast cancer, diabetes, neuropathy, and peripheral vascular disease has been highlighted [2]. In a similar manner, IRT has been successfully used in detecting skin cancer [3,4], monitoring skin burns [5], detecting problems associated with rheumatoid arthritis [6,7], detecting necrotizing enterocolitis [8], evaluating infectious diseases (such as the coronavirus 2019 (COVID-19)), and detecting fever conditions by working together with algorithms based on artificial intelligence [9,10], among others.

Regarding the use of IRT for skin cancer detection, the performance of the active/dynamic IRT stands out over passive IRT [11]. Buzug et al. [12] evidenced differences in thermoregulation curves between healthy and cancerous skin areas. Çetingül and Herman [13] proposed a methodology that quantifies the difference between the thermal responses of healthy skin and melanoma. Di Carlo et al. [14] showed that active thermography, unlike dermoscopy, distinguishes a clear pattern to differentiate basal cell carcinoma (BCC) tumors from actinic keratosis (AK). Godoy et al. [3] proposed a standardized analysis of dynamic thermography to discriminate malignant from benign lesions; the study included more than 100 patients, who presented benign, BCC, squamous cell carcinoma (SCC), or malignant melanoma (MM) lesions. A more sophisticated analysis of dynamic thermography is proposed by Godoy et al. [4]; it combines the thermoregulation curves (TRCs) modeling and a detection theory scheme, achieving a sensitivity and specificity of 99%. Magalhaes et al. [15] performed support vector machine classifiers to distinguish benign lesions (melanocytic nevi) from malignant melanoma lesions; this study explored features extracted from steady state (passive thermography) and dynamic thermography using a frequency of one frame per minute; thus, the steady state features were more relevant to solve this problem. Magalhaes et al. [16] proposed a deep learning classifier for processing passive thermography, achieving an accuracy of 96.91% in discriminating between malignant and benign lesions. However, when differentiating malignant from benign lesions, its performance declined considerably, highlighting the potential of active thermography to address the challenge of increased inter-class variability. Soto and Godoy [17] proposed key features and a scheme for skin cancer detection using active thermography and machine learning. Using a support vector machine (SVM) with a radial basis function (RBF) kernel classifier resulted in an accuracy close to 85%. Unlike other studies, Bu et al. [18] used active thermography with warm excitation to propose a 3D model of heat evolution in skin tumors; the simulations revealed a dependency between tumor thickness and the maximum contrast parameter, which serves as a discriminant in tumor classification. Similarly, using a cold stimulus, Cardoso and Azevedo [19] proposed a 3D model to analyze breast tumor sizes, considerably improving the contrast with the proposed methodology.

Despite decades of research and development of medical applications based on IRT, its massification has been limited mainly by the high cost of infrared (IR) cameras. According to Narayanamurthy et al. [20], IRT needs optimum instrumentation for recording purposes, given that it is widely affected by external noise. Due to the continuous development of electronic technology, new detector array structures and new semiconductor alloys are available. This has reduced costs and sizes and increased the resolution and precision of IR devices [21,22]. The development of longwave infrared (LWIR) microbolometer technology has led to the creation of multipurpose, low-cost cameras compatible with smartphones [23], and several other applications.

Recently, there has been high interest in the development of medical applications based on low-cost IR cameras, with tests being conducted mainly to support the evaluation of diabetic foot conditions [24,25,26,27,28,29,30,31]. Studies have evaluated skin burns [32] and assessed the healing progress of thoracic surgical incisions [33]. Villa et al. [28] characterized and compared the low-cost cameras Seek Thermal CompactPRO and Thermal Expert TE-Q1 Plus (TE-Q1), using the high-end camera INO IRXCAM-640 as a reference. Their findings suggest that TE-Q1 is suitable for e-health applications, particularly when assessing diabetic foot ulcers. Due to the quality of the measured noise equivalent temperature difference (NETD), the residual non-uniformity (RNU) validated at 25 °C is comparable to the IRXCAM-640 camera.

To determine if a camera is useful for a task, its performance should be evaluated by acquiring a dataset with said camera, the data should be processed, and its performance should be calculated. Medical problems involve requesting permission from the clinical unit, coordinating work teams, looking for volunteers, and preparing informative documentation for volunteers, among other things. Given that space and time are limited in a clinical environment, it is relevant to know the feasibility of using such equipment before going to any clinical trial. Moreover, most investigations are initially performed with high-quality IR cameras, so it is relevant to properly model the decrease in performance with lower-quality imagers. In this paper, we propose an IR video degradation model that degrades videos captured with high-quality cameras to simulate the performance of lower-quality cameras. Our case study evaluates the feasibility of using Xenics Gobi-640, Opgal Therm-App, and Seek Thermal CompactPRO cameras for skin cancer detection through active thermography. This evaluation is based on patient data previously acquired with a higher-quality imager [4]. However, the proposed degradation model can be applied to any kind of infrared camera, in any type of application in both passive and active thermography.

## 2. Theoretical Background

### 2.1. Skin Cancer

The skin is a large organ, covering approximately 16% of body mass. It is organized into two primary layers—the epidermis and dermis. The epidermis, which is considered the body’s armor, is a peripheral layer of skin that interacts with the environment and works as a physiochemical barrier against environmental stressors such as pathogens, chemicals, and ultraviolet (UV) radiation. The dermis, originating from the mesoderm, underlies the epidermis and serves as the anchorage for cutaneous structures such as hair follicles, nerves, sebaceous glands, and sweat glands. The dermis also contains abundant immune cells and fibroblasts, which actively participate in many physiological responses of the skin [34].

Thermal regulation is a relevant skin function used to maintain core temperatures within a small range, between 36 and 38 °C. Vasodilatation and vasoconstriction are among the main factors that control thermal regulation. Vasodilatation and vasoconstriction refer to changes in the blood vessel diameter, which affect skin temperature by changing the rate of blood exchange with the interior. If the ambient temperature rises, then increased conductance below the skin surface (due to increased blood flow) facilitates heat transfer from the body interior to the skin. In the cold, muscle tensing and shivering increase heat production and body temperature. As blood flow decreases, thermal conductance also decreases, decreasing heat loss from the body to the environment [35].

Skin—as any other organ—may be affected by solid malignant tumors. Solid malignant tumors require a blood supply in order to grow larger than a few millimeters in diameter. Tumors induce the growth of new capillary blood vessels (a process called angiogenesis) by producing specific angiogenesis-promoting growth factors [36].

With the presence of these new blood vessels and the increase in blood supply, the thermal response of an area with a tumor changes with respect to the thermal response without a tumor. By thermal response, we mean the recovery of the skin temperature when a stimulus is applied. Under this fundament, many studies have highlighted the potential of analyzing thermal recovery curves obtained from dynamic thermography in order to distinguish malignant from benign lesions [3,4,12,13,37].

### 2.2. Active Thermography

IRT is a fast, non-contact, and non-invasive technique used to analyze the temperature of an area of interest. It has been used to detect arthritis, allergies, breast cancer, and burns, among others. It allows a high-resolution view of the temperature of the lesion, from the epidermis to the depth of the tissue, in addition to measuring the variations in body temperature [2,38].

There are two types of IRT: passive and active. The passive IRT is used to evaluate the temperature of an object in a steady state, while the active IRT is used to evaluate the transient state after a thermal stimulus [1]. Active IRT is considered more powerful than passive IRT because it allows obtaining quantitative information related to the thermal properties of the sample [37]. When active IRT is used, one can plot the temperature measured by each pixel with respect to time obtaining one TRC per pixel.

### 2.3. Microbolometer Technology

IR technology is divided according to the wavelength response range: 0.75–1.4 μm near IR (NIR), 1.4–3 μm short wave IR (SWIR), 3–8 μm medium wave IR (MWIR), 8–15 μm LWIR, and 15–1000 μm far IR (FIR) [39].

Due to skin emissivity being close to 1, it can be considered a blackbody. Based on Wien’s displacement law, it is possible to define the maximum radiation wavelength of the skin $\lambda_{max}$ [37]. Typically, the superficial skin temperature is 27 °C, so $\lambda_{max}$ is approximately 10 μm. Then, LWIR technology, which is the most appropriate for skin screening, is the most widely used [2].

The most popular LWIR commercial detectors are made from an alloy of $Hg_{1-x}Cd_{x}Te$, corresponding to Quantum Well and non-refrigerated bolometer technologies [40]. The Quantum well-infrared photodetector (QWIP) technology is preferred in research and development for its high thermal sensitivity (approximately 30 mK), higher temporal resolution (100 to 200 Hz), and higher spatial resolution (up to 1280 × 1024 pixels) [37]. However, cameras based on QWIP technology are refrigerated, which implies a high cost. As an alternative, non-refrigerated microbolometer technology has emerged, which today is the most popular, especially low-cost commercial versions [23].

Recently low-cost microbolometers have entered the market with lower sensitivity (about 70 mK), spatial resolution (about 320 × 240 pixels), and temporal resolution (8 to 25 Hz). To tackle the lower sensitivity of these cameras, Bonmarin et al. [41] proposed using dynamic thermography, specifically the lock-in thermal imaging technique. The lower spatial resolution should not be a problem because it can be improved by choosing a suitable optical setup. Concerning temporal resolution, according to [37], dermatological applications do not require high temporal resolution, because the conduction of heat through the human skin is relatively slow (with respect to metals, for example). Based on these fundamentals, we propose using low-cost microbolometer cameras for medical applications, specifically related to skin cancer.

## 3. Materials and Methods

This section describes the equipment and dataset used, detailing the capture process. We explain the proposed data adaptation scheme and a skin cancer detection scheme proposed by Soto and Godoy [17]. Combining both schemes evaluates the feasibility of using microbolometer cameras for skin cancer detection.

### 3.1. Infrared Imagers

This study utilized data captured by a high-quality IR camera, QmagiQ [42], which uses quantum dots in a well (DWELL) technology. The data were adapted in terms of thermal, spatial, and temporal resolutions to the characteristics of the following microbolometer cameras: Xenics Gobi-640 [43], Opgal Therm-App [44], and Seek Thermal CompactPRO [45]. Table 1 summarizes the technical features of these cameras. All cameras have a similar spectral range; QmagiQ presents the best features in terms of field of view (FOV) and temporal resolution and NETD, followed by Xenics Gobi-640, Seek Thermal CompactPRO, and Opgal Therm-App. The focal plane array (FPA) size of the QmagiQ camera is inferior to that of the other cameras.

### 3.2. Dataset

The dataset is composed of 144 videos acquired by S. E. Godoy in his doctoral research [46] with the support of the University of New Mexico’s Dermatology Clinic staff. The data were captured with informed consent, which precludes their public release and ensures the confidentiality of patient information through an identification system accessible only to medical staff nurses. The study mainly included Caucasian and Hispanic populations from New Mexico, with participants aged between 26 and 96 years, with the main inclusion criterion being that the patient had a lesion suspected of being skin cancer, such that the lesion was diagnosed via a biopsy. In summary, the dataset is composed of 87 lesions diagnosed as benign and 57 as malignant. Of the malignant lesions, 41 were diagnosed as BCC, 9 as SCC, and 7 as MM.

The data acquisition process and its basic processing for subsequent analysis are described below.

#### 3.2.1. Data Acquisition Process

The data acquisition was carried out using active thermography in a controlled environment, with room temperature in the range of 20 °C to 22 °C. The acquisition procedure consisted of selecting a region of interest (ROI) with a plastic marker, as shown in Figure 1. We then took a visible image and a 15 s IR sequence of the ROI, used as a reference. Subsequently, using an air conditioning unit, the skin registered in the ROI was cooled for 30 s. After the cooling stage, the skin was heated via thermoregulation to room temperature. During the cooling and thermal recovery stages, a 1.5 min IR sequence was captured.

#### 3.2.2. Image Registration Algorithm

For the correct analysis of the acquired images, it is necessary to apply an image registration algorithm, which is responsible for correcting involuntary movements of the patient, and aligning the data sequence. Along with this, it is useful to differentiate the areas of lesion (pigmented area) and healthy skin on IR images.

The registration algorithm used corresponds to the one presented in Díaz et al. [47], which consists of the following stages:Manually select the corners of the plastic marker in the visible image and the first IR image. For subsequent images in the IR sequence, the corners are automatically detected using the selections from the previous image as references.Estimate an affine transformation matrix that maps motion between consecutive images (one matrix is estimated for each pair of images).Apply the inverse transformation to each image to align the image sequence relative to the first IR image.

Once the image registration process is finished, a 3D array $u\in R^{I\times J\times K}$ is obtained, where I and J represent the number of horizontal and vertical pixels, respectively, within the ROI. K denotes the number of images that compose the array (1 visible and K-1 IR images). In what follows, we utilize the term $u_{k}$ to denote the *k*-th frame within the video sequence, i.e., $u_{k}=u\left(\cdot ,\cdot ,k\right)\in R^{I\times J}$. Additionally, we refer to the (i,j)-th pixel within the *k*-th frame as $u_{i,j,k}$.

This registration algorithm provides a maximum image shift of three pixels. Because of this, once the data cube is registered, a 3-pixel border is removed to avoid processing TRCs with temperature measurements of the plastic marker. A set of nearby TRCs is highly correlated, so in general, this does not affect the data processing thanks to the TRC selection stage of the detection algorithm explained in Section 3.3. It is possible that with poorer-quality images, the image motion may be greater. However, this can be corrected with digital image processing techniques such as filter smoothing and contrast adjustment.

### 3.3. Skin Cancer Screening

To evaluate the effect of using lower-quality IR videos in skin cancer detection, we used the machine learning algorithm proposed in a previous work [17]. As shown in Figure 2, the detection scheme is composed of six stages, which are described as follows:Lesion selection. A mask is created over the visible image to delineate the lesion area, generated manually by outlining the pigmented region. Consequently, two sets of TRCs are established: one designated as *L*, comprising TRCs within the lesion area delineated by the mask, and the other denoted as *N*, consisting of TRCs within the non-lesion area.Initial temperature estimation. The subsequent step involves the selection of TRCs, which depends on the initial temperature of each TRC. In order to select curves whose initial temperature is less affected by non-uniformities in the cooling process, each TRC is modeled using a double exponential function, defined as follows:
(1)$$f_{i,j}\left(t\right)=\theta_{i,j}^{\left(1\right)}+\theta_{i,j}^{\left(2\right)}\exp\left[\theta_{i,j}^{\left(3\right)}t\right]+\theta_{i,j}^{\left(4\right)}\exp\left[\theta_{i,j}^{\left(5\right)}t\right],$$
whose parameters $\theta_{i,j}=\left[\theta_{i,j}^{\left(1\right)}\;\cdots \;\theta_{i,j}^{\left(5\right)}\right]$ are computed with a nonlinear least squares fitting, such that $f_{i,j}\left(kT_{s}\right)=u_{i,j,k}$. Here, $T_{s}$ is defined by the inverse of the camera frames per second (fps). With this model, the initial temperature is estimated by simply setting t=0 in (Equation 1).TRCs selection. A reference temperature is calculated as $T_{ref}=E\left[f_{L}\left(0\right)\right]$, where *E* is the expectation operator and $f_{L}\left(0\right)$ is the vector function of the initial temperature of the TRCs of the modeled lesion area within the *L* set.The selection of points to process considers a margin of error of p·100% with respect to $T_{ref}$. In this way, the set of points to be used is defined as follows:
(2)$$S=\left\{\left(i,j\right):\left|f_{i,j}\left(0\right)-T_{ref}\right|\leq p\cdot T_{ref}\right\},$$
in this case, we experimentally define p=0.01 as a good value. Using the set *S*, the sets of TRCs with similar initial temperatures from the lesion and non-lesion areas are defined as $L^{*}=L\cap S$ and $N^{*}=N\cap S$, respectively.Representative TRCs. For each set $L^{*}$ and $N^{*}$, a representative TRC is computed as the average TRCs among the selected pixels within each set, generating the average curves $\bar{TRC_{L^{*}}}$ and $\bar{TRC_{N^{*}}}$.Feature extraction. From the representative TRCs, a combination of features is extracted (features vector). In this case, the feature vector is as follows:$\left[E_{d},\sigma_{\rho_{B-L}},\sigma_{proj_{B-N}},\sigma_{d_{M-N}}\right]$, which are detailed in Section 3.3.1.Classifier. As the final step, the feature vector is processed by a classifier, which defines if the lesion is suspicious or not. In this study, we evaluated the performance of several machine learning techniques, including K-nearest neighbors (KNN), SVM with RBF kernel, random forest, and eXtreme Gradient Boosting (XGBoost). The random forest classifier achieved the best results, as detailed in Section 4.2. The results for the other classification techniques are provided in Appendix B.

#### 3.3.1. Feature Extraction Techniques

From the representative TRC of the lesion ($\bar{TRC_{L^{*}}}$) area and non-lesion area ($\bar{TRC_{N^{*}}}$), the following features are extracted:Euclidean distance (*d*). This feature is calculated as the norm of the difference $\bar{TRC_{L^{*}}}-\bar{TRC_{N^{*}}}$, normalized by the amount of points, i.e., as shown in the following equation:
(3)$$d=\frac{1}{K-1}∥\bar{TRC_{L^{*}}}-\bar{TRC_{N^{*}}}∥.$$The concept behind this feature is as follows: A small Euclidean distance indicates similar curves, making it highly likely that the lesion is benign. Conversely, a large Euclidean distance indicates significant differences in thermal recovery between the curves. This suggests that the lesion is likely malignant, as its thermal recovery behavior deviates from that of normal tissue [3].Energy difference ($E_{d}$). Let $\bar{\bar{TRC_{X^{*}}}}=\bar{TRC_{X^{*}}}-min\left(\bar{TRC_{X^{*}}}\right)$, i.e., the unbiased TRC of the $X^{*}$ area. The energy difference $E_{d}$ is calculated as follows:
(4)$$E_{d}=\left|∥\bar{\bar{TRC_{L^{*}}}}{∥}^{2}-{∥\bar{\bar{TRC_{N^{*}}}}∥}^{2}\right|.$$This feature is closely related to *d*. However, because of the triangular inequality $E_{d}\leq d$, $E_{d}$ quantifies smaller differences than *d*.Statistical similitude features. Here, six base features are defined to measure the similarity of a set of TRCs to a normalized modeled TRC, using the dual exponential model shown in (Equation 1). It is assumed here that the modeled TRC has *K* sample points and, thus, $f_{m}\in R^{K}$. With this, its inner product is computed by $<f,g>=f^{T}g$, where $f,g\in R^{K}$, and ^T^ denotes the column-vector transpose. $f_{m}$ is obtained by computing the five model parameters $\theta_{i,j}$ using a non-linear least-squares fitting approach. These parameters are averaged to obtain the *descriptive TRC*$f_{M}$ for each class (namely, cancerous TRCs and non-cancerous TRCs). Then, $f_{m}$ is forced to have a unit norm, i.e., $∥f_{m}∥≜\sqrt{<f_{m},f_{m}>}=1$.Let $f_{m}$ and $f_{n}$ denote a model TRC and an arbitrary TRC, respectively, both modeled and normalized (here, it is assumed that $∥f_{m}∥=1$). The following characteristics of projection, correlation, and Euclidean distance are described below.
(a)Projection. The projection of $f_{n}$ onto $f_{m}$ is calculated as $proj_{n}=<f_{n},f_{m}>$. This operation is performed for a set of at least 10 TRCs, and then the mean projection, $\bar{proj}$, and the standard deviation of the projection, $\sigma_{proj}$, are calculated over this set of projections.(b)Correlation. The correlation of $f_{n}$ onto $f_{m}$ is calculated using the following expression:
(5)$$\rho =\frac{cov(f_{m},f_{n})}{\sigma_{f_{m}}\sigma_{f_{n}}},$$
where $cov(f_{m},f_{n})$ is the covariance between $f_{m}$ and $f_{n}$, and $\sigma_{f_{m}}$ and $\sigma_{f_{n}}$ are the standard deviations (in time samples) of $f_{m}$ and $f_{n}$, respectively. This operation is performed for a set of at least 10 TRCs, and then the $\bar{\rho }$ (the mean correlation) and $\sigma_{\rho }$ (the standard deviation of the correlation) are calculated.(c)Distance. This parameter is calculated according to (Equation 3) but using $f_{m}$ as a reference and a family of at least 10 TRCs, $f_{n}$. The mean value $\bar{d}$ and standard deviation $\sigma_{d}$ are calculated over this set of distance values.Considering a model of malignant and benign TRCs, $f_{M}$ and $f_{B}$, respectively, for each data cube, the projection characteristics $\bar{proj}$ and $\sigma_{proj}$, correlations $\bar{\rho }$ and $\sigma_{\rho }$, and distances $\bar{d}$ and $\sigma_{d}$ are calculated for the sets of TRCs $L^{*}$ and $N^{*}$. In this way, 24 features are extracted, and grouped into four categories based on the origin of the model curve and the analysis group. For example, the malignant model $f_{M}$ over non-lesion area $N^{*}$ allows us to extract $\bar{proj_{M-N}}$, $\sigma_{proj_{M-N}}$, $\bar{\rho_{M-N}}$, $\rho_{proj_{M-N}}$, $\bar{d_{M-N}}$, and $\sigma_{d_{M-N}}$ features.

### 3.4. Proposed Data Adaptation Process

The proposed data adaptation scheme was designed to adapt IR videos acquired with a high-quality imager, e.g., as described in Section 3.2, to the characteristics of lower-quality cameras. In this study, we intend to simulate the data acquisition with Xenics Gobi-640, Opgal Therm-App, and Seek Thermal CompactPRO, but the proposed approach can be utilized in any pair of cameras. In order to define the scheme, the temporal, spatial, and temperature resolutions of the cameras were analyzed.

In Figure 3, the proposed model for IR video degradation is presented, which, as mentioned earlier, consists of the following three edges involving five stages:Temporal resolution. This edge only contemplates the temporary downsampling stage. As the temporal resolution of the camera to adapt is lower than the camera that captures the data, the sequence of IR images is downsampled following the proportion $fps_{HQ}:fps_{cam}$, where $fps_{HQ}$ and $fps_{cam}$ denote the sample rate of the high-quality camera and the camera to adapt, respectively.Spatial resolution. This aspect addresses two areas: spatial resolution and optical distortion.
(a)Spatial downsampling. The QmagiQ camera has a high instantaneous field of view (IFOV); however, the size of its FPA is smaller than the FPA of the cameras to model, in terms of the number of pixels. To avoid introducing artificial elements with unknown effects, the spatial dimensions of the images were not modified.(b)Point spread function (PSF). Images captured by each camera are usually affected by blurring due to optical distortions of the lens of each camera. The point spread function (PSF) applied corresponds to a 2D model, which was calculated as described by Jara et al. [48]. This model considers both optical and electronic aberrations.In order to transfer the optical response of the camera to be simulated to the high-quality videos, the PSF obtained from the lower-quality camera is applied to each frame of the higher-quality camera. Thus, the modified frame is the result of the spatial convolution of the image $u_{k}$ with the PSF of the camera to simulate ${\mathrm{PSF}}_{\mathrm{cam}}$:
(6)$${\tilde{u}}_{k}=u_{k}*{\mathrm{PSF}}_{\mathrm{cam}}.$$It is important to mention that usually the PSF is used to correct the optical distortions. In this study, it is used to degrade the images; such a worst-case scenario is to be evaluated. So it is expected that using the real camera can improve the performance of the detection algorithm by correcting the optical distortion.Thermal resolution. In order to simulate the thermal sensitivity of a camera, adding the characteristic noise of the camera to be modeled is proposed.The IR images contain spatial and temporal noise. According to Feng et al. [49], the noise affecting images captured by microbolometer IR cameras contains low-, medium-, and high-frequency spatial noise, along with horizontal and vertical component noise and non-uniformities.For applications utilizing active thermography, temporal noise is equally or more important than spatial noise affecting the camera.
(a)Spatial noise. The spatial noise characteristics of the camera to be simulated are added according to the following expression: ${\tilde{u}}_{k}={\tilde{u}}_{k}+\beta$, where β is the spatial noise resulting from blackbody radiator measurements as described in Section 3.5.1. This spatial noise already contains the low-, medium-, and high-frequency components proposed by Feng et al. [49] since we extracted them from actual measurements.(b)Temporal noise. The temporal noise is composed of two components: low frequency ($N^{LF}$) and high frequency ($N^{HF}$). Where the low-frequency noise is associated with ambient temperature fluctuations, while the high-frequency noise is related to the chemical and electrical characteristics of each camera component.Therefore, the response of each detector over time is degraded according to the following expression:
(7)$${\tilde{u}}_{i,j,k}^{cam}={\tilde{u}}_{i,j,k}+N_{i,j,k}^{LF}+N_{i,j,k}^{HF}.$$To reiterate, the subscripts i,j,k mean that we are evaluating the noise component at the (i,j)-th pixel, at the *k*-th frame.

### 3.5. Noise Characterization of Imagers

#### 3.5.1. Spatial Noise Characterization

As described earlier, the spatial noise of microbolometer IR cameras consists of low-, medium-, and high-frequency spatial noise, horizontal and vertical component noises, and non-uniformities. To transfer these noise features, a characteristic noise β of the camera to be modeled was extracted from measurements on a blackbody radiator.

To extract spatial noise, a sequence of 180 frames was captured. In our study, we considered setting the room temperature to 20 °C because the clinical data acquisition protocol stipulates that the ambient temperature should be in the range of 20 °C to 22 °C [3]. Each camera was positioned in front of the blackbody, ensuring that the entire width of it was visible so that all cameras covered the same field of view at the same time.

Using the cube u, an image $\bar{u}$ was generated by averaging all the 180 frames. This effectively removed the temporal noise present in each frame, leaving only the spatial noise components. Then, the spatial noise of a camera, β, is defined as follows:

$\bar{u}$ minus its two-dimensional mean value. Thus, β contains the horizontal, vertical, high, medium, and low-frequency noises, as well as the non-uniformities of each camera to be simulated, as we previously discussed.

In the study, for each camera, β was calculated for six measurements on the blackbody, i.e., between 15 °C and 40 °C with a step of 5 °C. The temperature range was selected to model the camera noise within the body temperature ranges (recall that we want to assess the feasibility of these cameras in medical applications).

#### 3.5.2. Temporal Noise Characterization

Usually, applications utilizing active thermography require a controlled temperature environment. In our skin cancer detection application, according to the data acquisition protocol, the room was set to a temperature between 20 °C and 22 °C using an air conditioning (AC) unit [3]. In our laboratory experiments with these cameras, we observed that due to the on–off control performed by the AC, fluctuations in ambient temperature affected the camera measurements.

To understand and model the ambient temperature fluctuations generated by the AC affecting the cameras to be simulated, measurements were taken using a Mikron M345 blackbody (manufactured by LumaSense Technologies, Inc., Santa Clara, CA, USA), with the Xenics Gobi-640, Opgal Therm-App, and Seek Thermal CompactPRO microbolometer cameras. All these elements were placed in a 5 m^2^ room, which was conditioned with an AC unit of 10,000 BTU. The ambient temperature fluctuations were measured using an Elitech RC-4HC thermometer, manufactured by Elitech Technology, Inc. (Milpitas, CA, USA).

Similar to the spatial noise characterization, measurements were conducted by stabilizing the blackbody between 15 °C and 40 °C with a 5 °C step. These measurements were performed by setting the ambient temperature of the AC to 15 °C, 20 °C, and 25 °C.

In Figure 4a–c, a sample of measurements over 5 min on the blackbody with the Xenics, Opgal, and Seek cameras, respectively, is presented. It can be observed that the Xenics and Seek cameras exhibit jumps, which are effects of the Non-Uniformity Correction (NUC). It is also noticeable that all cameras exhibit low-frequency oscillatory behavior related to ambient temperature fluctuations. For the Opgal camera, a highly correlated behavior with ambient temperature fluctuations is observed, demonstrating a linear relationship during periods of increasing or decreasing ambient temperatures. Changes in ambient temperature are reflected in the camera’s response with a variable delay, and this response is attenuated.

On the other hand, the Xenics and Seek cameras exhibited fluctuations in their measurements over time; their behaviors were not easy to comprehend because the NUC modified the gain of the measurements to correct errors originating from changes in ambient temperature. Applications utilizing active thermography ideally require smoothness over time in the temperature measurements of each detector in a camera. Therefore, jump correction produced by the NUC was applied to the Xenics and Seek cameras, as shown in Figure 5 and Figure 6, respectively.

According to these observations, it is not possible to accurately model low-frequency temporal noise as a function dependent on ambient temperature variations. This is because the NUC makes the camera’s corrective actions unpredictable. However, low-frequency oscillations are observed, which we suggest modeling using a Fourier series for each detector, according to the following equation:$$N_{i,j,k}^{LF}=\sum_{z=0}^{Z}a_{i,j}^{z}cos\left(z\omega_{i,j}\cdot \frac{k}{f_{s}}\right)+b_{i,j}^{z}sin\left(z\omega_{i,j}\cdot \frac{k}{f_{s}}\right).$$

Then, the high-frequency component $N_{i,j}^{HF}$ corresponds to white Gaussian noise with standard deviation $\sigma_{i,j}$, whose value is calculated over the difference between the measurement over time of the detector u(i,j) and its low-frequency component $N_{i,j}^{LF}$.

## 4. Results

Two main experiments were conducted and are shown here. The first aims to validate the proposed degradation model, and the second evaluates the feasibility of using low-cost IR cameras in skin cancer detection using active thermography.

### 4.1. Validation of the Degradation Model

To validate the degradation model, a simultaneous video was acquired with the three cameras positioned over a lesion on a volunteer. The video was captured according to the protocol described in Section 3.2.1, and then each video was registered using the registration algorithm described in Section 3.2.2.

To validate the model, the video acquired with the Xenics Gobi-640 camera was considered a high-quality video. Thus, the videos captured with the Opgal Therm-App and Seek Thermal CompactPRO cameras, which have lower quality and costs, will be mimicked. Following the methodology described in Section 3.4, the video captured with the Xenics camera was adapted to the noise characteristics of the other cameras. First, the frame rate and image dimensions were reduced to match those of the camera being modeled. Second, the PSF characteristics of each camera were applied to the reduced-sized imagery. Third, characteristic spatial noise was added to the degraded imagery. Fourth, temporal noise specific to the camera being modeled was also added.

Thus, let $u_{i,j,\cdot }$ be a TRC captured at position (i,j); · indicates that we are considering all time samples of the video. The temporal noise present in $u_{i,j,\cdot }$ is $NT_{i,j}=u_{i,j}-f_{i,j}$, where $f_{i,j}$ corresponds to the curve $u_{i,j}$ modeled as a double exponential, as defined in (Equation 1). Then, the low-frequency component is obtained by modeling $NT_{i,j}$ as a Fourier series, as defined in Section 3.5.2, to obtain $N_{i,j}^{LF}$. Meanwhile, the high-frequency component corresponds to white Gaussian noise, whose standard deviation is calculated over the residual noise, as described above.

Now, with the data cube $u^{c}$ acquired using camera *c* and the data cube ${\tilde{u}}^{c}$ from a high-quality camera degraded to mimic camera *c*, the Pearson correlation coefficient $\rho (u_{i,j}^{c},{\tilde{u}}_{i,j}^{c})$ was calculated to estimate the level of similarity between the actual and simulated TRCs. Mimicking the Opgal Therm-App camera, which included 6889 TRCs, achieved an average correlation of 0.9756±0.0104. Meanwhile, mimicking the Seek Thermal CompactPRO camera, which included 7031 TRCs, achieved an average correlation of 0.9271±0.0861.

Given that the average Pearson correlation coefficient exceeded 0.9, the correlation between the real and simulated curves was deemed very strong [50]; this indicates a high degree of similarity between the data collected with the actual camera and the data that mimicked it. Therefore, we consider the proposed degradation model to be valid within the scope of the TRC measurements. Clearly, more evidence must be collected to validate our model to mimic more cameras and different case studies.

A sample of the degradation model applied to the data cube acquired with the Xenics camera to Opgal camera characteristics is presented in Figure 7, where Figure 7a corresponds to an image of the IR video acquired with the Xenics camera, and Figure 7d corresponds to a TRC of the cube. Figure 7b corresponds to an image from the video acquired with the Opgal camera, and Figure 7e corresponds to an image of the cube. Figure 7c corresponds to an image of the video acquired with the Xenics camera that was modified to features of the Opgal camera according to the proposed model, and Figure 7f corresponds to an image of the degraded cube.

Similarly, a sample of the degradation model applied to the same camera, modified to mimic the features of the Seek camera, is presented in Figure 8. Figure 8a corresponds to an image of the IR video acquired with the Xenics camera, and Figure 8d corresponds to the TRC of the cube. Figure 8b corresponds to an image from the video acquired with the Seek camera, and Figure 8e corresponds to an image of the cube. Figure 8c corresponds to an image of the video acquired with the Xenics camera that was modified to features of the Seek camera according to the proposed model, and Figure 8f corresponds to an image of the degraded cube.

The proposed method does not rely on mimicking the TRCs exactly at every sample point, but to transfer all the noise characteristics from one camera to the other. Even though the TRCs are not visually identical, we observe that the low-frequency oscillations and high-frequency noise from the low-cost cameras are included in the modeled data (see Figure 8e,f, for example). Some of the differences that are not transferred are due to the NUC correction shutter that the cameras have constantly operating.

A larger graphic sample is presented in Appendix A. Where the results of mimicking each camera on the four types of lesions studied in this work are shown.

### 4.2. Feasibility Study of Using Low-Cost IR Cameras in Skin Cancer Detection

We now utilize the degraded data cube to assess the performance of the skin cancer detection algorithm we describe in Section 3.3. As such, we can evaluate the performance one may obtain when low-cost infrared imagers can achieve this task.

To understand the variability of the classifier performance, each one of the modeled datasets was trained using the bootstrap method, which involved taking multiple samples with replacements from the original dataset and generating a new dataset of the same size. The modified dataset was then divided, allocating 80% of the data for training and 20% for testing. This process was repeated 2000 times. The results were previously reported by our group.

The algorithm performance with the original dataset acquired with the QmagiQ camera is shown in the second column of Table 2. The results obtained by degrading these videos to match the characteristics of the Xenics, Opgal, and Seek cameras are presented in the following columns of the same table. The reported indices include accuracy, true positive rate (TPR), true negative rate (TNR), and positive predictive value (PPV), indicating the minimum, maximum, average (AVG), and standard deviation (SD) values for the evaluation.

As expected, the best performance is achieved with the QmagiQ camera, with an average accuracy of 87.29%, sensitivity (TPR) of 87.26%, specificity of 87.15%, and precision (PPV) of 87.39%. The adaptations to the Xenics and Opgal cameras offer a similar performance to that of the QmagiQ camera. The Xenics camera reached an accuracy of 84.33% and a sensitivity of 83.03%, while the Opgal camera achieved an accuracy of 84.20% and a sensitivity of 83.23%. Based on these results, both the Xenics and Opgal cameras are suitable for skin cancer detection, as they achieve similar levels of performance to the QmagiQ camera, with approximately a 3% difference in accuracy.

The worst performance was observed when adapting to the characteristics of the Seek camera, with an average accuracy of 82.13%, sensitivity of 79.77%, and specificity (TNR) of 83.74%. The sensitivity was approximately 8% lower than that of the QmagiQ camera. Consequently, this camera is not really suitable for the skin cancer application we are investigating.

Regarding the system’s performance in detecting the different types of skin cancer, the highest sensitivity was obtained in detecting SCC lesions, with values of 87.66%, 90.92%, 94.74%, and 95.94%, with the Seek, Opgal, Xenics, and QmagiQ cameras, respectively. Concerning BCC lesions, sensitivity values of 78.68%, 83.60%, 83.18%, and 87.23% were achieved with the Seek, Opgal, Xenics, and QmagiQ cameras, respectively. The worst performance was obtained in detecting MM lesions, with sensitivity values of 74.98%, 71.68%, 66.91%, and 76.42%, with the Seek, Opgal, Xenics, and QmagiQ cameras, respectively.

Table 3 presents the performances in terms of the sensitivity and specificity of the proposed methods using different cameras, along with the performances of highly trained dermatologists using naked eye evaluations and dermoscopy. The results obtained by Magalhaes et al. [16] using passive thermography and deep learning are also presented. Dermoscopy significantly outperformed naked eye evaluation, demonstrating very high performance. However, it is important to note that this level of performance is achieved by dermatologists with years of clinical training. Along with this, the reported performances correspond to different detection problems. The passive thermography analysis method using deep learning proposed by Magalhaes et al. [16] outperforms dermatologists in this detection problem. However, when distinguishing between malignant and benign lesions (with the malignant class including MM, BCC, and SCC), the algorithm’s performance drops considerably. This is where the relevance of active thermography analysis becomes evident, achieving over 80% sensitivity and 85% specificity. This work demonstrates that it is possible to significantly reduce equipment costs while maintaining similar performance.

## 5. Discussion and Conclusions

In this work, a degradation model of IR videos is proposed and evaluated to mimic the performances of different cameras in medical applications under laboratory conditions. Based on this model, videos captured with a high-quality camera were degraded to the characteristics of three low-cost imagers, namely, the Xenics Gobi-640, Opgal Therm-App, and Seek Thermal CompactPRO cameras. These synthetic datasets were then used to evaluate the feasibility of using the modeled cameras over the skin cancer detection algorithm proposed by Soto and Godoy [17].

The proposed degradation model focuses on three key areas to accurately mimic the performance of any camera: temporal, spatial, and thermal resolution. It has been demonstrated that the model achieves a high level of similarity in the TRC, which is the most important aspect in applications that use active thermography. Moreover, qualitatively, it is appreciated that, spatially, the model manages to transmit the texture of the images captured by the modeled cameras.

The proposed model may not fully transfer the characteristics of a lower-quality camera to images captured with a higher-quality camera. The main characteristics that the model does not consider are as follows:Adjustment of the image size. The model does not adjust the size of high-quality images to match those captured with lower-quality cameras when the FPA of the higher-quality camera is larger. This approach was not considered because it requires an interpolation process, which may introduce noise that is not characteristic of the camera being simulated.Shape and size of the detector. This feature is critical for determining the minimum size of detectable objects. However, since the size of skin lesions is significantly larger than the detector size, this characteristic was not considered. The size of a typical detector in microbolometer technology is approximately 20 μm; with the right optics, it would be possible to detect a 40 × 40 μm lesion.Temporal noise introduced by the shutter. When analyzing the temporal variations in the cameras due to changes in ambient temperature, the camera adjusts the offset and modifies the gain. However, incorporating this feature is problematic because the camera’s logic for applying these adjustments is unknown and cannot be determined.

Despite not considering the points mentioned above, the proposed model achieves a high similarity between the degraded videos and the original ones. We anticipate that when these cameras are used in a clinical environment, their behavior will mirror what was described in this study. This is because the cameras were evaluated in one of the worst scenarios in terms of ambient temperature fluctuations. The air-conditioning unit aggressively controlled the room temperature, and when activated, it decreased the temperature by approximately 5 °C, which translated into a drift in the temperature measurements. However, we do not rule out the possibility that the camera measurements might be influenced by other issues that we have not yet considered.

We believe that this model can be useful to evaluate the use of an IR camera in different applications, without the need to spend time, space, and other resources generating a dataset with a camera that does not fit the requirements of the application. However, as we have shown, IR cameras are highly affected by the ambient temperature, so it is important to characterize the behavior of the camera in the environment in which it will be used so that the simulation is as close as possible to its real performance. Nevertheless, some nuances may not be captured by our model, and that is exactly what we are currently doing in our current research.

We showed that the model manages to simulate the behaviors of the three cameras studied, achieving a high similarity of TRCs, with a Pearson correlation coefficient higher than 0.9. We believe that this model can be applied to most microbolometer technology chambers. However, we cannot be sure, especially in cameras with worse characteristics than those studied.

For our case study, we demonstrated that—within certain boundaries—the Xenics Gobi-640 and Opgal Therm-App are the most suitable IR cameras for skin cancer detection using active thermography. This is because their characteristics allow the evaluated algorithm to achieve similar performance to the high-quality camera. Moreover, the NUC approach in these two cameras is less aggressive, allowing discontinuities to be easily corrected over time. Meanwhile, the Seek camera presents an average decline of 5% in accuracy and 7% in TPR, so we do not consider it suitable for skin cancer detection using active thermography.

It is relevant to analyze the performance of the tool in detecting different types of skin cancer to observe its robustness in different scenarios. The performance of the system is outstanding when processing SCC-type lesions, reaching an average sensitivity between 87.66% and 95.94%; with BCC-type lesions, an average sensitivity between 78.68% and 87.23% is achieved. Whereas, when analyzing MM lesions, the average sensitivity is in the range of 66.91% to 76.42%. It is important to note that the dataset used is small, with only seven cases of MM, which implies that the tests performed with the bootstrap technique contain at most two IR cubes of MM lesions. This is a disadvantage at the time of training the detection models because it does not manage to represent the generality of the behavior of this type of lesion; and at the time of evaluation, it implies that when a case is missed, the sensitivity is reduced to 50%. Because of this, it is very important to increase the dataset to make the system more robust.

This study was conducted using a dataset captured from the New Mexican population, primarily involving Caucasian and Hispanic individuals. Although we consider the dataset small, it is representative of this demographic. We strongly believe that the malignancy of the lesions is encrypted in the thermal recovery of the skin, so we consider it important to increase the dataset to ensure it represents all types of populations. We are currently working to conduct a similar study in the Chilean population of the Biobío region. Conducting a clinical study involves many challenges, starting with the authorization of the clinical field, which must ensure the confidentiality of the patient’s data and that the patient is not exposed to any treatment that is harmful to his or her physical and mental health. For this study, we obtained informed consent from all subjects, and the clinical staff was responsible for anonymizing the data, ensuring that we could never correlate patient identities with their data. Another challenge is adjusting to the limited space and time to capture the data in a real public health clinical setting. This was a main motivator for developing the video degradation model, which allowed us to select the right equipment for the research. This model reduced the time required to use multiple devices simultaneously and improved our ability to work comfortably in constrained spaces.

Most of the algorithms attempt to find a good approach to detect skin cancer with the best-available sensor; our research attempts to change the paradigm by evaluating the feasibility of these algorithms when applied to data acquired with low-cost sensors. The similarity of the results between QmagiQ, Xenics, and Opgal cameras is due to the robustness of the detection algorithm. As evidenced by the results presented in Section 4.2 and Appendix B, more advanced classification techniques such as random forest, SVM, and XGBoost allow for similar performance in skin cancer detection when using high- or low-quality technology, as opposed to simpler algorithms, such as KNN. Thus, we can conclude that—thanks to advances in machine learning and feature extraction techniques—it is possible to utilize lower-quality technology. We hope to report better results soon; we are developing new statistical tools to extract the spatial thermal information of a process that is hidden within noisy TRCs. For this, the degradation model will play a key role in understanding the ways the actual data are acquired with low-cost imagers.

As with any other type of cancer, early detection is key for skin cancer patients. The tool presented in this work uses non-invasive, non-contact, and low-cost technology that can screen a suspicious lesion within a few minutes. The portability and low cost of our tool allow its rapid massification, allowing it to be accessible in primary care centers, even in difficult-to-access villages, where it is complex for a patient to be evaluated by a trained dermatologist. Our tool also aims to support specialists in whether to perform a biopsy or not; with this, we hope to contribute to the optimization of resources, avoiding unnecessary biopsies. this way, our tool contributes to public health by aiding in the early detection of skin cancer. It serves as an initial screening to refer patients to a dermatologist, supports the decision to perform a biopsy, and helps reduce the costs associated with more severe diseases caused by the late detection of cancerous lesions.

## Figures and Tables

**Figure 1 sensors-24-05152-f001:**
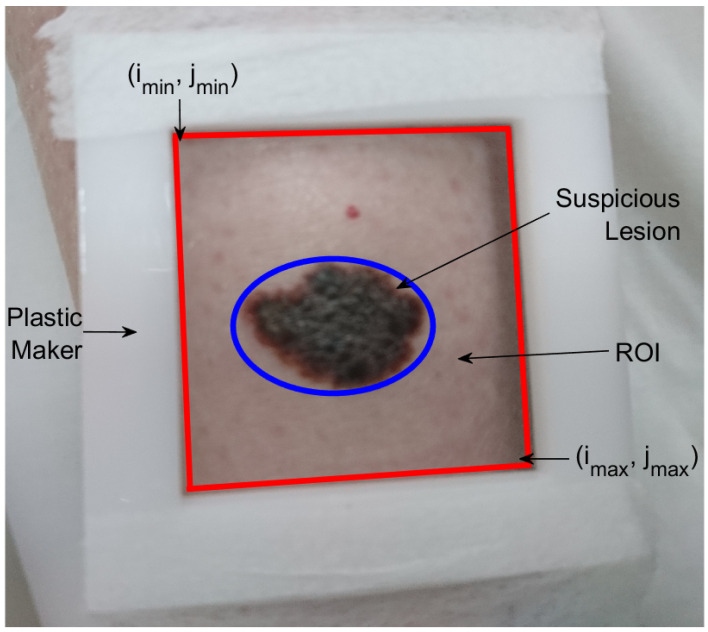
Example of a plastic marker used to select the region of interest. The region of interest is indicated in red, and the suspicious lesion is highlighted in blue.

**Figure 2 sensors-24-05152-f002:**
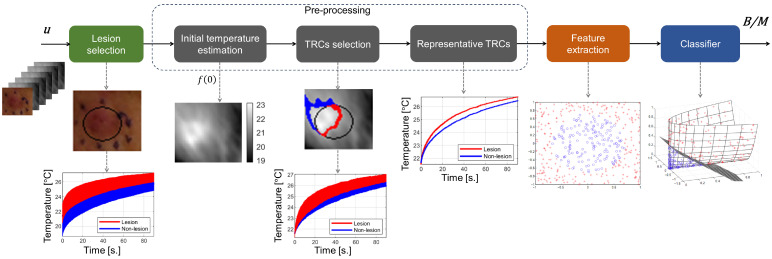
Detection scheme used to evaluate the feasibility of using different IR cameras in detecting skin cancer using active thermography.

**Figure 3 sensors-24-05152-f003:**
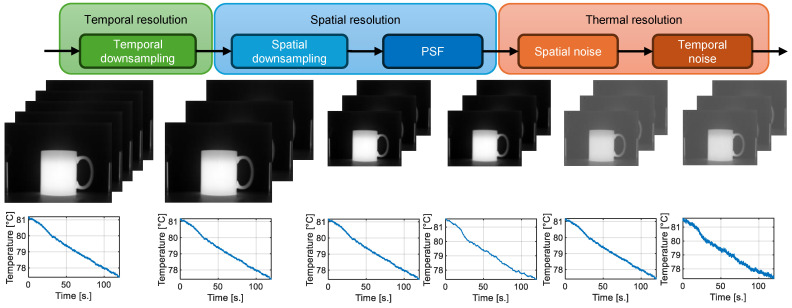
Schematic of the proposed degradation model, which addresses 3 areas: temporal, spatial, and thermal resolution. Giving rise to a process composed of 5 stages.

**Figure 4 sensors-24-05152-f004:**
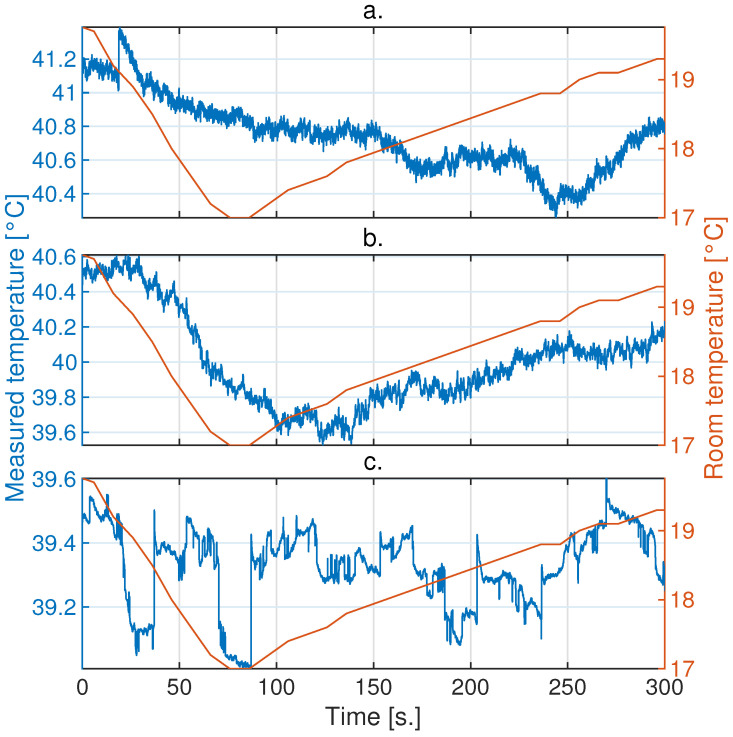
Samples of the measurement behaviors from different IR cameras on a blackbody stabilized at 40 °C in a room with controlled ambient temperature at 20 °C using an AC unit. The order of measurements taken with the different cameras is as follows: (**a**) Xenics Gobi-640, (**b**) Opgal Therm-App and (**c**) Seek Thermal CompactPRO.

**Figure 5 sensors-24-05152-f005:**
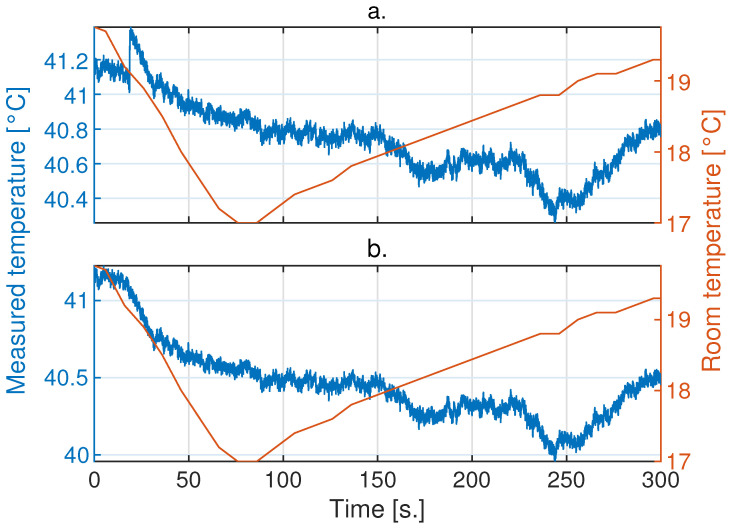
Sample of the results of the jump correction produced by the NUC in the Xenics Gobi-640 camera. (**a**) Uncorrected measurements; (**b**) corrected measurements.

**Figure 6 sensors-24-05152-f006:**
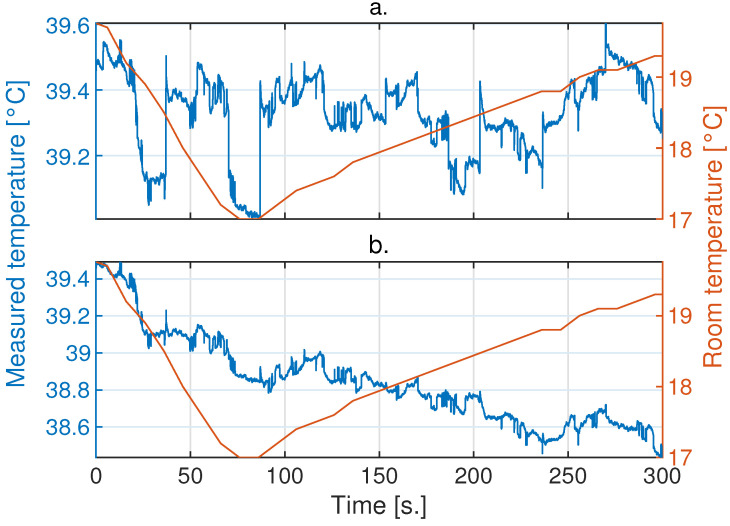
Sample of the results of the jump correction produced by the NUC in the Seek Thermal CompactPRO camera. (**a**) Uncorrected measurements; (**b**) corrected measurements.

**Figure 7 sensors-24-05152-f007:**
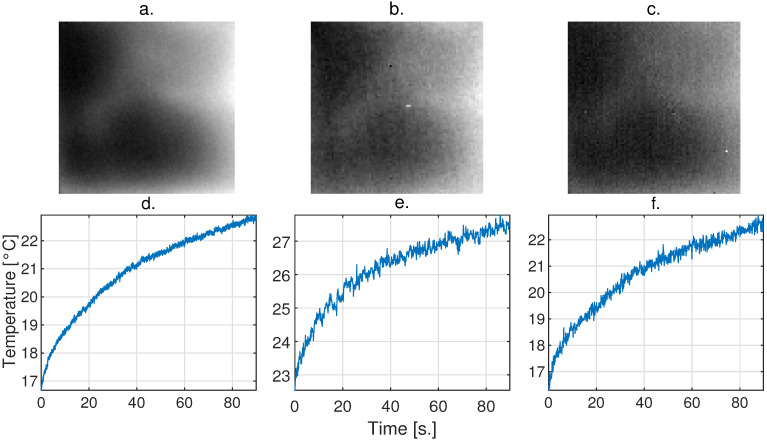
Sample of the degradation performed on a high-quality video to mimic Opgal Therm-App camera features. (**a**) Image captured with the Xenics Gobi-640 camera, (**b**) image captured at the same instant of time and same area with the Opgal Therm-App camera, (**c**) Xenics image adapted to Opgal camera features, (**d**,**e**) correspond to representative TRCs of the video acquired with the Xenics and Opgal cameras, respectively. (**f**) TRC from the adaptation.

**Figure 8 sensors-24-05152-f008:**
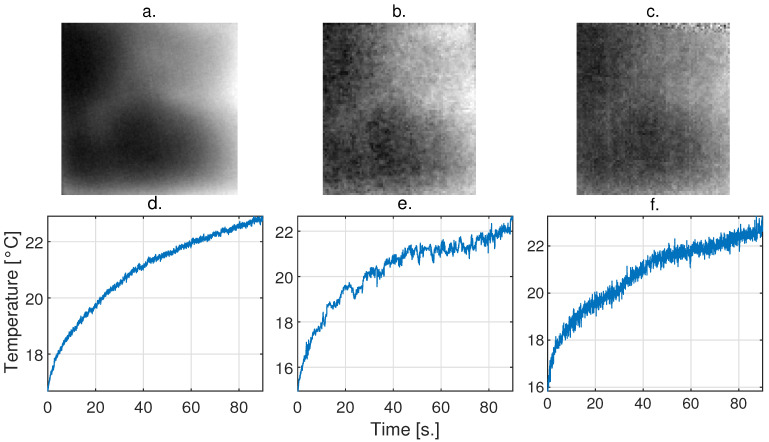
Sample of the degradation performed on a high-quality video to mimic Seek Thermal CompactPRO camera features. (**a**) Image captured with the Xenics Gobi-640 camera, (**b**) image captured at the same instant of time and same area with the Seek Thermal CompactPRO camera, (**c**) Xenics image adapted to Seek camera features, (**d**,**e**) correspond to representative TRCs of the video acquired with the Xenics and Seek cameras, respectively. (**f**) TRC from the adaptation.

**Table 1 sensors-24-05152-t001:** Technical features of the involved cameras.

	Camera	QmagiQ	Xenics Gobi-640	Opgal Therm-App	Seek Thermal Compact PRO
Characteristic					
Focal distance [mm]		50	25	6.8	12.5
Spectral range [μm]		8–14	8–14	7.5–14	7.5–14
FOV [°]		5.5 × 4.4	25.84 × 19.38	55.56 × 41.67	33.40 × 24.81
Temporal resolution [fps]		60	50	8.7	15
FPA size [pixels]		320 × 256	640 × 480	384 × 288	320 × 240
NETD [mK]		20	50	70	70
Manufacturer		QmagiQ, LLC (Nashua, NH, USA)	Xenics nv (Leuven, Belgium)	Opgal Optronic Industries Ltd. (Karmiel, Israel)	Seek Thermal Inc. (Santa Barbara, CA, USA)

**Table 2 sensors-24-05152-t002:** Performance of the skin cancer detection algorithm with a random forest classifier, processing data captured with a high-quality QmagiQ camera and adapted to features of the Xenics, Opgal, and Seek cameras.

Camera Adaptation	QmagiQ (Original)			Xenics			Opgal			Seek		
**Index**	**Min.**	**Max.**	**AVG ± SD**	**Min.**	**Max.**	**AVG ± SD**	**Min.**	**Max.**	**AVG ± SD**	**Min.**	**Max.**	**AVG ± SD**
Accuracy (%)	62.07	100.00	87.29 ± 6.60	51.72	100.00	84.33 ± 7.11	55.17	100.00	84.20 ± 7.09	51.72	100.00	82.13 ± 7.43
TPR (%)	33.33	100.00	87.26 ± 11.49	28.57	100.00	83.03 ± 12.62	27.27	100.00	83.23 ± 12.58	28.57	100.00	79.77 ± 13.79
TPR MM (%)	0.00	100.00	76.42 ± 36.29	0.00	100.00	66.91 ± 40.11	0.00	100.00	71.68 ± 38.14	0.00	100.00	74.98 ± 37.30
TPR BCC (%)	20.00	100.00	87.23 ± 13.60	20.00	100.00	83.18 ± 14.96	0.00	100.00	83.60 ± 14.63	16.67	100.00	78.68 ± 16.40
TPR SCC (%)	0.00	100.00	95.94 ± 15.81	0.00	100.00	94.74 ± 18.37	0.00	100.00	90.92 ± 24.09	0.00	100.00	87.66 ± 26.62
TNR (%)	47.62	100.00	87.39 ± 8.98	47.62	100.00	85.28 ± 9.50	44.44	100.00	84.94 ± 9.75	36.84	100.00	83.74 ± 10.23
PPV (%)	28.57	100.00	82.45 ± 11.94	28.57	100.00	79.11 ± 12.90	25.00	100.00	78.91 ± 12.74	26.67	100.00	76.96 ± 13.25

**Table 3 sensors-24-05152-t003:** Performances of classical skin cancer detection methods and automatic passive and active thermography methods.

Methodology	Detection Problem	TPR (%)	TNR (%)
Naked eye evaluation [51]	MM vs. benign	71.00	81.00
Dermoscopy evaluation [51]	MM vs. benign	90.00	90.00
Passive thermography + deep learning [16]	MM vs. benign	94.12	98.41
Passive thermography + deep learning [16]	Malignant vs. benign	62.81	57.58
Active thermography − QmagiQ camera	Malignant vs. benign	87.26	87.39
Active thermography − Xenics camera	Malignant vs. benign	83.03	85.28
Active thermography − Opgal camera	Malignant vs. benign	83.23	84.94
Active thermography − Seek camera	Malignant vs. benign	79.77	83.74

## Data Availability

The dataset used is private due to the confidentiality with which it was acquired.

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
