# Peer review of "Feasibility Study on the Use of Infrared Cameras for Skin Cancer Detection under a Proposed Data Degradation Model"

_sensors, 2024, doi:10.3390/s24165152_

Round 1

Reviewer 1 Report

Comments and Suggestions for Authors The manuscript titled "Feasibility study of the use of infrared cameras for skin cancer detection under a proposed data degradation model" presents a proposal for a degradation model for videos acquired with high-quality IR cameras to mimic the performance of a lower-quality camera.   The authors clearly presented the procedure for image degradation, as well as the performance results of the algorithm for detecting skin cancer from high and low quality images.   Below are some considerations that need to be clarified:   1 - Why not present the effect of degradation of images and temperature curves in some cases of skin cancer?   2 - Why not present a comparative study between degraded images and real images obtained by cameras with similar characteristics?   3 - What other characteristics of a lower quality camera were not considered in the proposed model? What is the limitation for including these points?   4 - The authors could compare the skin cancer estimation results obtained after image degradation with research that used lower quality cameras.   5 - Consider adding to the state of the art other works that evaluated the use of dynamic thermal images in tumor detection, such as:   - https://doi.org/10.1016/j.cmpb.2023.107562 - https://doi.org/10.3390/s22093327

Reviewer 2 Report

Comments and Suggestions for Authors

1.      While the study provides useful insights, it also has restrictions that might affect the applicability and reliability of the results in real-world scenarios. Please discuss all of these limitations in the discussion:

a.      The study was done in a controlled setting with specific temperatures. In real life, conditions vary, and this might affect the results. It may not show how well the low-cost cameras work in different real-world environments.

b.      It is a small sample size study. The study used data from a limited number of volunteers. This small sample might not represent the larger population accurately.

c.      The model simulates low-cost cameras by degrading high-quality images. Simulated data might not capture all the real-life issues and nuances of using actual low-cost cameras.

d.      The study only tested three specific low-cost cameras. Results might not apply to other low-cost infrared cameras.

e.      There is an algorithm dependency. The success of the study depends heavily on the performance of the skin cancer detection algorithm. If the algorithm is not robust, the results might not be reliable or generalizable.

f.       Data privacy and ethical considerations were managed but still present challenges. handling real patient data requires strict ethical standards, which can be complex.

2.      The study did not directly compare infrared thermography results to traditional methods like dermoscopy. Including a comparison would provide a clearer understanding of the effectiveness of this new method. The authors must describe how the results of infrared thermography compare to traditional methods like dermoscopy in terms of sensitivity and specificity. My suggestion, see this example of a reference that was not included, and use it to answer and enhance your paper: Infrared Macrothermoscopy Patterns - New Category of Dermoscopy. J Imaging. 2023 https://pubmed.ncbi.nlm.nih.gov/36826955/

3.      Was the point spread function (PSF) used in the degradation model validated against actual optical distortions seen in the low-cost cameras? The PSF was applied but not explicitly validated against actual optical distortions. Validating the PSF with real distortions can improve the accuracy of the degradation model.

4.      Did you use cross-validation techniques to ensure the generalizability of the skin cancer detection algorithm? The study used the bootstrap method but did not mention cross-validation. Using cross-validation could help ensure the algorithm's performance is generalizable to new data.

5.      How would the use of low-cost infrared cameras impact cancer screening programs in low-resource settings? The study discusses feasibility but not the broader public health impact. Including this important medical discussion can highlight the potential benefits and challenges in hospital and clinical with low-resource settings.

6.      Were the thermal images evaluated for different types of skin cancer (e.g., melanoma, basal cell carcinoma) to see if accuracy varies? The study did not differentiate between types of skin cancer. Evaluating different types could provide more detailed insights into the technology's effectiveness to the readers. Dermatologists and medical doctors must know this point. Talking about skin cancer is simply very generic and not very useful to oncology.

7.      Were the image registration techniques sufficient to correct all patient movement artifacts? How are you sure about it? The study used an image registration algorithm but did not quantify how well it corrected all artifacts. Providing this information would clarify the quality of the medical thermal images.

8.      Were there any considerations for patient consent and data privacy specifically mentioned in the study? I did not see it. The study mentions consent but does not detail data privacy measures. Including these details would address ethical concerns comprehensively.

9.      Did you test the algorithm's performance with different machine learning models to compare results? The study used a support vector machine (SVM) with an RBF kernel but did not test other models. Testing multiple models could ensure the best performance is achieved and makes the study more reliable for medical use.

10.   Was there any analysis on the minimum detectable lesion size using the low-cost cameras? This is extremely important to the medical doctors. The study did not mention the minimum detectable lesion size. Including this analysis would show the practical limitations of the low-cost cameras.

11.   Were any potential confounders like patient skin type or environmental factors considered in the analysis? This is a mandatory item in any imaging study in dermatology. The study controlled the environment but did not mention patient skin types or other potential confounders. Considering these factors would strengthen the study's validity.

12.   For a long time, the skin is not the largest organ of the body, but rather the microcirculatory system. This must be corrected in the text. The study would benefit from supervision by a medical doctor with knowledge in the field of thermography to assist researchers in future studies.

Round 2

Reviewer 1 Report

Comments and Suggestions for Authors

The authors were able to clarify all questions about the manuscript.

Author Response

We appreciate your comments, they were very important to improve the quality of the article.

Reviewer 2 Report

Comments and Suggestions for Authors

Thank you for your detailed response. It is important to remember that your justifications and explanations should be included in your manuscript, not just in your response to the reviewers. These explanations are crucial for your readers to fully understand the limitations and potential of your study. Your response must clarify and inform the readers. So it should be incorporated into your manuscript text. This will provide scientific rigor and transparency, allowing readers to fully grasp the limitations and scope of your study and increase the scientific perception.

Discussing the broader public health impact of using low-cost infrared cameras is important. You need to highlight the potential benefits and challenges in low-resource settings. Including this in your manuscript will show the medical importance of your study, not just the engineering perspective.

Also, include an analysis or at least a discussion on the performance of your tool for different types of skin cancer, even if your current dataset is small and not medical supervised. Explain the importance of this differentiation. Quantifying how well your image registration algorithm corrected patient movement artifacts should be added to clarify the quality of the medical thermal images. These limitations are important to describe.

Providing a comparison of different machine learning models in your manuscript will show that you have explored all options to achieve the best performance as you explained me. Discussing potential confounders such as patient skin type and environmental factors in your manuscript will strengthen the validity of your study. In no way does it detract from the results, this will show more depth and academic maturity of the authors. 

The study you considered relevant to highlight the relevance of your work that was not included is also an example of a high-cost sensor to reinforce the importance of low-cost cameras. It will make the impact of your work even greater, as it will prove to those who criticize that low-cost devices do not allow tumor diagnosis, and that only high-cost sensors should be used. This is something important that can demystify a misconception. You have this opportunity in this study.

Making these changes is important to give more weight and relevance to your research, ensuring it is well-received and respected in the scientific community. We look forward to receiving the revised manuscript with the necessary adjustments and corrections.
